# Study protocol for Hear Me Read (HMR): A prospective clinical trial assessing a digital storybook intervention for young children who are deaf or hard of hearing

Chenelle Miller[1]☯, Kelly M. Boone[1]☯, Prasanth Pattisapu[2,3,4‡], Prashant Malhotra[3,4‡]*

**1** Center for Biobehavioral Health, Abigail Wexner Research Institute at Nationwide Children's Hospital, Columbus, Ohio, United States of America, **2** Center for Child Health Equity and Outcomes Research, Abigail Wexner Research Institute at Nationwide Children's Hospital, Columbus, Ohio, United States of America, **3** The Ohio State University College of Medicine, Columbus, Ohio, United States of America, **4** Nationwide Children's Hospital, Columbus, Ohio, United States of America

☯ These authors contributed equally to this work.
‡ PP and PM also contributed equally to this work.
* Prashant.Malhotra@nationwidechildrens.org

**Data Availability Statement:** No datasets were generated or analysed during the current study.

## Abstract

Since the early 2000's, digital reading applications have enhanced the language and literacy skills of typically hearing young children; however, no digital storybook intervention currently exists to scaffold the early language and literacy skills of their peers who are deaf or hard of hearing. To address this gap, our research team developed a novel digital storybook intervention called Hear Me Read with the aim of enhancing the therapeutic, language, and literacy benefits of speech-language therapy. This prospective clinical trial (registered at clinicaltrials.gov, NCT#: 05245799) aims to determine the efficacy of adding Hear Me Read to in-person speech-language therapy for children aged three to five years who are deaf or hard of hearing. Fifty caregivers, their child, and their child's treating speech-language pathologist participate in the trial for 12 months. In the first six months, children attend standard-of-care speech-language therapy sessions. In the second six months, children continue to attend standard-of-care speech-language therapy sessions *and* use the Hear Me Read application, via a study supplied iPad. The primary outcome of this trial is that, compared to in-person speech-language therapy alone, in-person speech-language therapy with Hear Me Read will *improve vocabulary, speech, and language* outcomes in children aged three to five years who are deaf or hard of hearing. The secondary outcome is that, compared to in-person speech-language therapy alone, in-person speech-language therapy with Hear Me Read will improve *literacy* outcomes in children aged three to five years who are deaf or hard of hearing. The goal of this intervention is to help children who are deaf or hard of hearing achieve their vocabulary, speech, language, and literacy goals through interactive digital storybook reading.

**Funding:** This research was supported, in part, by the National Center for Advancing Translational Sciences of the National Institutes of Health under Grant Number UM1TR004548. Funds from Grant Number UM1TR004548 awarded to the Ohio State University's Center for Clinical and Translational Science and internal support from The Abigail Wexner Research Institute at Nationwide Children's Hospital. This information or content and conclusions are those of the authors and should not be construed as the official position or policy of, nor should any endorsements be inferred by the U.S. Government. Funds were awarded to PM. The funder provided support for conducting the research but did not have any role in the study design, data collection and analysis, decision to publish, or preparation of the manuscript. The specific roles of these authors are articulated in the 'author contributions' section. There was no additional external funding received for this study.

**Competing interests:** I have read the journal's policy and the authors of this manuscript have the following competing interests: Dr. Malhotra developed a company, Digital Story Therapies, Inc, based on the technology inherent in the Hear Me Read application. A financial conflict of interest was identified. Immediately upon its identification, Dr. Malhotra created a Conflict Management Plan with his institution, halted role as Principal Investigator, and ceased all study related duties. These duties were transferred to Dr. Prasanth Pattisapu, the current Principal Investigator. Digital Story Therapies, Inc has no relationship or influence with the conduct of this clinical trial [See amended funding statement in cover letter].

## Introduction

Hearing loss affects as many as 5 per 1,000 children aged 3 to 17 years [1, 2]. Children who are deaf or hard of hearing (D/HH) face challenges in spoken language, literacy, vocabulary, and their overall reading comprehension [3–6]. Poor language and literacy skills are associated with poor socio-emotional development and academic achievement [7, 8]. Therefore, an intervention to support language and literacy development for children who are D/HH is needed [9].

Prior research suggests that early intervention, technological advancements (e.g., hearing aids, cochlear implants, mHealth), and speech-language therapy (in-person or telehealth) support the language and literacy development of children who are D/HH [2, 10–12]. Nonetheless, children who are D/HH are reported to be at least one standard deviation behind their typically hearing peers on standardized language assessments [13]. This outcome is not completely ameliorated by technology such as cochlear implants, as children who are D/HH demonstrate lower vocabulary knowledge than their typically hearing peers [14]. Reading presents a powerful mechanism from which children develop language and literacy skills. Current speech-language therapy approaches often utilize storybook reading techniques to facilitate language and vocabulary development. With the influx of digital education tools and technology [15], electronic books and storybook applications present an opportunity to enhance the language and literacy outcomes of children who are D/HH.

Electronic storybook interventions for children who are D/HH are novel. Only two studies previously investigated the outcomes of using E-books on signed and spoken vocabulary acquisition for children aged 2 to 9-years who are D/HH. Results showed improved vocabulary in the enhanced E-book intervention conditions (i.e., E-books with signed support, response-contingent feedback, phonology highlights, embedded word definitions, and direct instruction of the targeted words) compared to control E-books (i.e., E-books without enhanced vocabulary supports) [16, 17]. While both studies demonstrate vocabulary gains, neither holistically looks to improve the language *and* literacy outcomes of children who are D/HH. Prior literature with typically hearing children suggests that successful electronic storybooks that support collaborative reading and limit distractions, act as a supplemental reading intervention and impact literacy and language outcomes similar to that of traditional books. Despite this finding, there are no digital storybook applications tailored to support both language and literacy skills in children who are D/HH [18–22]. To address this gap, our research team developed a novel digital storybook application called Hear Me Read (HMR) to (1) extend the language and literacy benefits of speech-language therapy, (2) present language input using multiple modalities [19], (3) support collaboration between caregivers and speech-language pathologists (SLPs), and (4) facilitate shared reading between caregivers and their child.

This manuscript outlines the study protocol for the HMR prospective clinical trial, which aims to provide evidence on the efficacy of adding HMR to in-person speech-language therapy on the vocabulary, speech, language, and literacy outcomes of children, aged 3 to 5 years, who are D/HH. This study hypothesizes that in-person speech therapy with the HMR application will improve the vocabulary, speech, language, and literacy outcomes of children aged 3 to 5 years, who are D/HH, compared to in-person therapy alone.

## Materials and methods

### Ethics

The Institutional Review Board (IRB) at Nationwide Children's Hospital approved the study (STUDY00001863) on February 10[th], 2022 (See S2 File. Study Protocol; See S1 File. SPIRIT

Checklist for protocol-specific reporting guidelines). Research staff submit study-related changes (e.g., changes to study procedures, protocol, and eligibility criteria) to the IRB at Nationwide Children's Hospital.

All caregivers provide written informed consent (See S3 File. Study Consent Form) for their and their child's participation. The child's SLP provides verbal consent for their own participation, under a waiver of documentation of written consent. Research staff remind participants of their voluntary participation and their right to withdraw their participation. If a participant discontinues the intervention protocol, research staff explain that the participant can (1) completely withdraw from the intervention and all study procedures, (2) withdraw from the intervention and allow research staff to collect outcome measures, or (3) if willing, after any concerns are addressed, continue participation in both the intervention and study procedures.

This prospective clinical trial was registered on ClinicalTrials.gov (NCT05245799) on February 18th, 2022, prior to enrollment of any participants. Anticipated completion of primary analysis is September 2024. Study information and results will be submitted to ClinicalTrials. gov, in line with their requirements.

## Study design

This single-site, prospective clinical trial is conducted at Nationwide Children's Hospital in Columbus, Ohio. This trial utilizes standard-of-care speech-language therapy assessments and recommendations to establish a baseline of how children who are D/HH perform in vocabulary, speech, language, and literacy domains. The study aims to understand how vocabulary, speech, language, and literacy outcomes change after implementing the HMR application.

Study participation lasts for approximately one year. During the first six months, children participate in speech-language therapy, per standard of care recommendations (SLT-only). During the second six months, participants continue to receive standard of care speech-language therapy, and *in addition*, use the HMR application (SLT+Digital). See Fig 1 for the schedule of enrollment. In-person therapy frequency is determined clinically by the SLP and may range from weekly to every 6-months, but frequency is expected to remain the same for the duration of the trial. Standardized reading time is recommended by the SLP, and prescribed at 20 minutes, 3 times per week for the duration of the trial.

The intervention aims to complement speech-language therapy; therefore, standard-of-care assessments and recommendations remain the same in the SLT-only and SLT+Digital phases of the study, with the only difference being the implementation of the HMR application. The sequential order of this within-subjects design eliminates potential carry-over effects that would occur if participants enrolled in the SLT+Digital then SLT-only intervention.

The study hypotheses are:

1. In-person speech-language therapy with a novel digital storybook intervention (HMR) **improves** vocabulary, speech and language outcomes in young children who are D/HH compared with in-person therapy alone.

2. In-person speech-language therapy with a novel digital storybook intervention (HMR) **improves** literacy outcomes in young children who are D/HH compared with in-person therapy alone.

## Data collection

Data is collected from three sources: child, caregiver, and SLP. Data collection occurs at three timepoints, at baseline (pre-trial), approximately six-months after enrollment (SLT-only), and approximately twelve months after enrollment (SLT+Digital).

| | STUDY PERIOD | | | |
|---|---|---|---|---|
| | Eligibility Screening | Pre-trial | SLT-Only | SLT+Digital |
| TIMEPOINT | $-t_1$ | 0 | $t_1$ | $t_2$ |
| ENROLLMENT: | | | | |
| Eligibility Screen | X | | | |
| Informed Consent | X | | | |
| Speech-Language Evaluation | X | ◀━━━━━━━━━━━━━━━━━▶ | | |
| INTERVENTIONS: | | | | |
| Speech-Language Therapy | X | ◀━━━━━━━━━━━━━━━━━▶ | | |
| Hear Me Read (HMR) | | | | X |
| ASSESSMENTS: | | | | |
| Clinical Evaluation of Language Fundamentals Preschool-3 (CELF-P3) | X | ◀━━━━━━━━━━━━━━━━━▶ | | |
| Receptive One-Word Vocabulary Picture Test (ROWVPT-4) | X | ◀━━━━━━━━━━━━━━━━━▶ | | |
| Pre-trial Survey (Caregiver) | | X | | |
| Post-SLT Survey (Caregiver and SLP) | | | X | |
| Post-SLT+Digital Survey (Caregiver and SLP) | | | | X |

**Fig 1. Schedule of enrollment.**

## Study endpoints

**Child assessments.** SLPs conduct routine speech-language evaluations every six months, per standard of care. Assessments include the Receptive One-Word Vocabulary Picture Test (ROWVPT-4) and subtests of the Clinical Evaluation of Language Fundamentals Preschool-3 (CELF-P3) (Refer to Table 1 for a detailed explanation of these assessments). The ROWVPT-4 and several subtests (i.e., Word Structure, Sentence Comprehension, Expressive Vocabulary, Following Directions, Recalling Sentences, Basic Concepts, and Word Classes) and indices (i.e., Core Language Score, Receptive Language Index, and Expressive Language Index) of the CELF-P3 measure the primary study endpoints of *vocabulary*, *speech*, *and language* outcomes of children who are D/HH. Additional subtests (i.e., Pre-literacy Rating Scale and Phonological Awareness) and indices (i.e., Early Literacy Index) of the CELF-P3 measure the secondary study endpoint of *literacy* outcomes of children who are D/HH (Refer to S1 Table for a detailed description of CELF-P3 subtests and indices). While enrolled in HMR, treating SLPs video record the child's speech-language evaluations. An independent SLP, who is blinded to the sequence of the assessment (i.e., SLT-only, SLT+Digital), reviews these recordings and acts as a reliability tester to ensure unbiased evaluation scores. Speech-language assessments are expected to remain the same for the SLT-only and SLT+Digital intervention periods; the independent SLP does not know if the video is of the child completing their SLT-only or SLT+Digital evaluation. There are no foreseen circumstances under which the outcomes accessor is unblinded.

**Caregiver-Reported measures.** Caregivers complete three surveys: *pre-trial*, *post-SLT-only*, and *post-SLT+Digital*, as well as weekly reading questionnaires. The *pre-trial survey* gathers information about the child (e.g., their race, speech and developmental delays, hearing diagnosis, and reading habits), the caregiver (e.g., their relationship to the child, age, marital status, level of education, mode of communication), and caregiver's experiences and expectations using digital reading applications. After the first 6 months, caregivers are asked to complete the *post-SLT-only survey* which collects caregiver-reported information on speech therapy frequency, home carryover, and assessment of the usefulness of storybook reading in therapy. At the end of the 12-month study, caregivers are asked to complete the *post-SLT+Digital survey* composed of open-ended questions about the HMR applications' usability and function. On a weekly basis, caregivers complete *reading questionnaires* assessing their child's

**Table 1. Description of the standardized assessments, ROWVPT-4 and CELF-P3, that measure the primary study endpoints (*vocabulary*, *speech*, *and language*) and secondary study endpoint (*literacy*).**

| Clinical Assessment | Description |
| --- | --- |
| Receptive One-Word Picture Vocabulary Test (ROWPVT-4) | 190-item developmentally conscious assessment of examinee's *receptive vocabulary*. The examiner tests examinee's ability to effectively match the examiner's spoken word to visual images of objects, actions, or concepts. Examiners present examinees with a subset of items–determined when the examinee makes several consecutive errors. Raw scores (0–100) are reported as standard scores, percentile ranks, scaled score, and age equivalents with higher scores indicating better performance [23, 24]. |
| Clinical Evaluation of Language Fundamentals Preschool-3 (CELF-P3) | An assessment of receptive and expressive language ability (*language and emergent literacy*). There are several subtests that target the receptive and expressive language of children ages 3–6 years 11 months. Each subtest yields a raw score reported as standard score, percentile rank, scaled score, and age equivalent with higher scores indicating better performance [23, 25]. (Refer to S1 Table for all relevant subtests and indices.) |

frequency of reading and the modality of reading (i.e., how much of their child's reading was done on a digital device).

**SLP-Reported measures.** SLPs complete two surveys for each participating child in their care: *post-SLT-only* and *post-SLT+Digital*. The *post-SLT-only survey* asks SLPs about speech therapy frequency, accordance with home-carryover, and their assessment of the usefulness of storybook reading in therapy. The *post-SLT+Digital survey* gathers information about SLPs perception of the HMR applications' applicability, useability, and feasibility.

**Sample adherence.** Research staff monitor in-app metrics and self-reported measures of participant book and reading engagement. The HMR application documents participants' progress in and comprehension of each book. Additionally, weekly reading questionnaires document caregiver-reported measures of time spent reading with their child during the week. Research staff follow-up with caregivers about book progress and incomplete weekly reading questionnaires monthly.

## Intervention

The IOS based HMR application was developed using the Unity operating system. Highlights for Children, Inc. provided access to high-quality illustrative children's books that were then imported into the HMR digital platform.

A dual interface, parent/child view and therapist view, allows for collaboration between treating SLPs and caregivers. In the "therapist view," the SLP assigns books based on the child's treatment goals (e.g., gaining proficiency in parts of speech, syllables, vowel sounds), reading level, the book's length, among other features. Once the SLP assigns a book, books appear on the "parent/child view." Here, caregivers can view personalized embedded prompts from the SLP, read books with illustrations turned on or off, enable the highlighting functions, and record (audio and video) themselves reading segments of a particular book for their child to view on their own.

A preliminary usability assessment of the HMR application identified that caregivers and children wanted an application that was easy to use (default needs), provided targeted literacy and language features (specific needs), and complemented their daily schedule (family needs). Considering these themes, the HMR application has a centralized layout allowing families to see all books assigned and navigate with ease. Additionally, the HMR applications' use of visual and auditory features provides the multimodal access to literacy and language that children who are D/HH benefit from. Furthermore, the recommendation for using the HMR application, 20 minutes per day, three times per work is in line with clinical recommendations allowing families to mimic their normal schedules when using the application [19]. See DeForte et al (2020) for a visual representation of the HMR application's multimodal functions. Lastly, caregivers are encouraged to utilize dialogic reading strategies–a shared-reading strategy found to positively affect children's literacy skills [26].

Overall, the HMR application provides a tailored and collaborative reading experience with the hope of working with speech-language therapy to target the vocabulary, speech, language, and literacy goals of children who are D/HH.

## Sample size calculation

Feasibility analysis of local patients indicated 411 children who are D/HH aged 3 to 5 years 11 months were seen in 2019. Of these, 141 children had a Speech-Language evaluation billed in that year. Our feasibility assessment motivates us to recruit a total of 50 children who are D/HH, their caregiver, and their SLP. This number accounts for eligibility screening, patient volume, and participant attrition. A sample size of n = 27 will provide 90% power to detect a

mean of paired differences of 0.75 with an estimated d = 1.0 and alpha = 0.001 using a two-sided paired t-test, so 50 participants will be sufficient to power the desired tests [27].

## Setting and participants

Participants are recruited from the Hearing Program at Nationwide Children's Hospital (Columbus, OH). The Hearing Program is a multidisciplinary team of professionals specializing in hearing loss (comprised of Pediatric Otolaryngology, Audiology and Speech-Language Therapy departments) that aims to help children who are D/HH achieve their full potential [28].

Research staff identify potentially eligible children via the electronic medical record and screen them for eligibility. Inclusion criteria are: (1) children between the ages of 3 and 5.11 years old at the time of the pre-trial assessment, (2) confirmed diagnosis of bilateral auditory neuropathy, bilateral sensorineural (SN) hearing loss, or bilateral mixed SN and conductive hearing loss, and (3) a pure tone average (PTA) score of greater than or equal to 30Db if diagnosed with bilateral SN hearing loss, or bilateral mixed SN and conductive hearing loss. Children are excluded if: (1) English is not the child's primary language, or (2) the child's Core Language Score (subtest of CELF-P3) on their pre-trial assessment is >2 standard deviations from the mean. Caregivers of eligible participants must be the participant's legal guardian, and a participating SLP must be treating the eligible participant.

## Recruitment

**Child and caregiver recruitment.** Research staff recruit eligible families in person (at outpatient speech-language pathology clinics) or virtually (via phone call or sending a recruitment letter via mail and/or email). During these interactions research staff provide additional information about the potential risks, benefits, commitments, and incentives associated with study participation. After reviewing the study components and answering any questions the caregiver may have, research staff ask the caregiver if they would like to participate in the study and guide the caregiver through the written informed consent process.

**Treating SLP recruitment.** Within Nationwide Children's Hospital there are six SLPs in the departments that make up the Hearing Program (Pediatric Otolaryngology, Audiology, and Speech-Language Therapy). All are invited to participate. Research staff recruit SLPs remotely (via phone call and/or send a recruitment letter via email) and give them information about the potential risks, benefits, commitments, and incentives associated with study participation. If an SLP is interested, research staff guide them through the verbal consent process and document the date of verbal consent. The IRB granted a waiver of written documentation of consent for SLPs.

## Data management

Electronic data capture, including electronic medical record (EMR) data abstraction and completion of caregiver/SLP surveys, takes place in REDCap, a secure, web-based software platform designed to support data capture for research studies [29, 30]. REDCap provides 1) an intuitive interface for validated data capture; 2) audit trails for tracking data manipulation and export procedures; 3) automated export procedures for seamless data downloads to common statistical packages; and 4) procedures for data integration and interoperability with external sources. Study staff manually abstract information from the EMR to determine eligibility. For each survey, caregivers and SLPs will receive a link that directs them to a secure REDCap survey, where they enter the information directly into the REDCap platform.

Research staff provide SLPs with an NCH study iPad to record the speech-language evaluations of patients enrolled in HMR. SLPs upload recordings to a secured NCH shared drive, then delete the recording from the iPad. When the HMR intervention is initiated, research staff provide caregivers with an NCH study iPad with the HMR application already installed, which collects and stores in-app metrics directly on the device. The following in-app metrics are collected and uploaded to the study database at the end of participation: participants time spent reading, their progress in the book (e.g., reading 6/6 pages), their utilization of audio-visual features (e.g., did the participant listen to the video recording, did they click on the underlined words to show accompanying pictures), and their comprehension score (e.g., did the participant understand the plot, characters, and setting). After transferring data from the iPads to a secured NCH shared drive, data is deleted from the iPads.

Research staff store deidentified data according to federal requirements, in a password protected database on an encrypted hospital server only accessible to study personnel. Participants are notified of their right to privacy and data safeguards during the informed consent process. Participants may opt to store their identifiable information, including their protected health information, for future research use and can withdraw their authorization to use or disclose their identifiable information at any time.

## Statistical analyses

The primary goal of the study is to compare the changes in vocabulary, speech, language, and literacy outcomes collected during the SLT-Only intervention period to the SLT+Digital intervention period. A repeated measures approach using a mixed effects model will be used to compare the change across these two intervention periods using a group by time interaction. For primary and secondary outcomes, changes in raw score, given its sensitivity, for the ROWPVT-4 and CELF-P3 will be evaluated. Raw scores are reported as standard scores, percentile ranks, growth scale, and age equivalents. Baseline assessments are used to identify patient characteristics (e.g., age, frequency of therapy, rapport with SLP) associated with outcome variables using t-tests and correlation analyses. Patient characteristics will be evaluated for their associations with changes in outcome. Characteristics associated with changes in the outcome (e.g., patient characteristics such as age, frequency of therapy) will be adjusted for within the context of the repeated measures approach using a mixed effects model. Interim analyses are not planned. Access to primary and secondary outcome analyses will be made available on ClinicalTrials.gov in line with their requirements. All statistical analyses will be conducted using SAS Enterprise Guide [31]. Outcomes are reported for the intention-to-treat population and for the per protocol population. A 2-sided $P < .05$ is considered to be statistically significant and clinically relevant.

## Discussion

While interventions exist to support the language and literacy development of children who are D/HH, this population continues to display prolonged language and literacy deficits, compared to their typically hearing peers [32–34]. Previous research suggests that digital technologies may foster literacy learning [35, 36]. Additionally, using dialogic reading strategies may improve oral language and vocabulary skills [37–39]. Coupled together, dialogic reading strategies and digital reading technologies present an opportunity to support the literacy and language development of children who are D/HH. However, current digital storybook applications are not sensitive to the developmental nuances of these children, and the literature examining digital reading interventions for this population is limited [19, 16, 17], highlighting the need for a digital reading experience like the HMR application.

In this manuscript, we detail a prospective clinical trial that aims to evaluate the efficacy of adding a digital reading intervention to speech-language therapy on the vocabulary, speech, language, and literacy outcomes of children who are D/HH. We hypothesize adding the HMR application to speech-language therapy will improve the vocabulary, speech, language, and literacy outcomes of children aged 3–5.11 years old who are D/HH. The study has clinical implications. The HMR application intertwines therapeutic goals (e.g., supporting language and literacy development) and reading such that the child's reading experiences to their clinical goals.

Our research team aims to add to the limited literature on digital storybook interventions for children who are D/HH by providing this population a novel reading experience unique to their multimodal needs. We anticipate that the HMR application will provide children who are D/HH a digital reading experience tailored for their needs. At the conclusion of this study, we will determine if adding the HMR application to speech therapy impacts the vocabulary, speech, language, and literacy outcomes of children who are D/HH.

## Supporting information

**S1 File. SPIRIT checklist.**
(DOCX)

**S2 File. Study protocol.**
(DOCX)

**S3 File. Study consent form.**
(DOCX)

**S1 Table. Subtest and indices for the CELF-P3 clinical standard-of-care assessment of receptive and expressive language and emergent literacy skills.**
(DOCX)

## Acknowledgments

We would like to thank "Highlights for Children", Jodilyn Butkovich, Tendy Chiang, Jillian Foutz, Kenisha Hicks, Janelle Huefner, Mary Lofreso, Shana Lucius, Jonathan Luna, Sophia Nichols, Anand Saytapriya, Lauren Wills, and Lauren Yoshihiro.

## Author Contributions

**Conceptualization:** Prashant Malhotra.

**Funding acquisition:** Prashant Malhotra.

**Investigation:** Prashant Malhotra.

**Methodology:** Prashant Malhotra.

**Project administration:** Chenelle Miller, Kelly M. Boone, Prasanth Pattisapu, Prashant Malhotra.

**Resources:** Chenelle Miller, Kelly M. Boone, Prasanth Pattisapu, Prashant Malhotra.

**Supervision:** Kelly M. Boone, Prasanth Pattisapu, Prashant Malhotra.

**Writing – original draft:** Chenelle Miller.

**Writing – review & editing:** Chenelle Miller, Kelly M. Boone, Prasanth Pattisapu, Prashant Malhotra.

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
