## [Decision Letter · Decision Letter 0]

7 Dec 2023

PONE-D-23-28173Study protocol for Hear Me Read (HMR): A prospective clinical trial assessing a digital storybook intervention for young children who are deaf or hard of hearingPLOS ONE

Dear Dr. Malhotra,

Thank you for submitting your manuscript to PLOS ONE. After careful consideration, we feel that it has merit but does not fully meet PLOS ONE’s publication criteria as it currently stands. Therefore, we invite you to submit a revised version of the manuscript that addresses the points raised during the review process.

We look forward to receiving your revised manuscript.

Kind regards,

Renato S. Melo, PhD

Academic Editor

PLOS ONE

Journal Requirements:

PSM 

This research was supported, in part, by the National Center for Advancing Translational Sciences of the National Institutes of Health under Grant Number UM1TR004548. Funds from Grant Number UM1TR004548 support the Ohio State University’s Center for Clinical and Translational Science. 

https://ccts.osu.edu/

https://www.nationwidechildrens.org/research/resources-infrastructure/core-facilities/center-for-clinical-and-translational-science-osu#:~:text=The%20Ohio%20State%20University%20Center,disease%20prevention%20strategies%20and%20the

Clinical and Translational Intramural Funding Program from the Abigail Wexner Research Institute, Grant AWD00001881

https://www.nationwidechildrens.org/research/clinical-research/resources-for-investigators

Funders reviewed and approved the proposed study design. Funders did not have any additional roles.

I have read the journal's policy and the authors of this manuscript have the following competing interests: Dr. Malhotra developed a company, Digital Story Therapies, Inc, based on the technology inherent in the Hear Me Read application. A financial conflict of interest was identified. Immediately upon its identification, Dr. Malhotra created a Conflict Management Plan with his institution, halted role as Principal Investigator, and ceased all study related duties. These duties were transferred to Dr. Prasanth Pattisapu, the current Principal Investigator. Digital Story Therapies, Inc has no relationship or influence with the conduct of this clinical trial.  

We note that one or more of the authors are employed by a commercial company: Digital Story Therapies, Inc

“The funder provided support in the form of salaries for authors, but did not have any additional role in the study design, data collection and analysis, decision to publish, or preparation of the manuscript. The specific roles of these authors are articulated in the ‘author contributions’ section.”

4. We noted in your submission details that a portion of your manuscript may have been presented or published elsewhere. Figure 2 is taken from "DeForte S, Sezgin E, Huefner J, Lucius S, Luna J, Satyapriya AA, Malhotra P. Usability of a Mobile app for improving literacy in children with hearing impairment: focus group study. JMIR Human Factors. 2020; 7(2), e16310. doi:10.2196/16310". This is not a dual publication since the manuscripts are about two distinct areas of the same study. The current manuscript is a study protocol paper while the other (cited) manuscript is about the development of the Hear Me Read application. Outside of this figure, there are no replications, in this manuscript, of the pervious publication. Please clarify whether this [conference proceeding or publication] was peer-reviewed and formally published. If this work was previously peer-reviewed and published, in the cover letter please provide the reason that this work does not constitute dual publication and should be included in the current manuscript.

Reviewers' comments:

Reviewer's Responses to Questions

**Comments to the Author**

1. Does the manuscript provide a valid rationale for the proposed study, with clearly identified and justified research questions?

Reviewer #1: Yes

Reviewer #2: Yes

Reviewer #3: Yes

2. Is the protocol technically sound and planned in a manner that will lead to a meaningful outcome and allow testing the stated hypotheses?

Reviewer #1: No

Reviewer #2: Yes

Reviewer #3: Yes

3. Is the methodology feasible and described in sufficient detail to allow the work to be replicable?

Reviewer #1: No

Reviewer #2: No

Reviewer #3: No

4. Have the authors described where all data underlying the findings will be made available when the study is complete?

Reviewer #1: No

Reviewer #2: No

Reviewer #3: No

5. Is the manuscript presented in an intelligible fashion and written in standard English?

Reviewer #1: Yes

Reviewer #2: Yes

Reviewer #3: Yes

6. Review Comments to the Author

You may also provide optional suggestions and comments to authors that they might find helpful in planning their study.

Reviewer #1: In this article, the authors have introduced a novel digital storybook intervention intended to enhance the effectiveness of standard speech-language therapy. To assess the impact of incorporating the digital storybook, known as "Hear Me Read," into the standard therapy, they have proposed a prospective clinical trial. The suggested approach involves enrolling individuals for a 12-month period, where they will receive standard care alone (SLT-only) during the first six months and then standard care combined with the Hear Me Read application (SLT+Digital) during the subsequent six months. Data will be collected at the end of each six-month phase. The authors anticipate that any disparity in outcomes between the two conditions (SLT-only vs. SLT+Digital) will reflect the additional benefit of the digital component.

A Major Concern:

This particular pre-post design is suitable when one assumes that the underlying mechanism, in this case, the improvement in children's speech and language skills, remains relatively consistent or stable over time. It is reasonable to assume that providing standard therapy over the 12-month period alone should lead to some level of improvement in the measured outcomes. If that is indeed the case, the difference between the two conditions (SLT-only during the first six months and SLT+Digital during the last six months) will encompass both the impact of 'natural progress' (i.e., improvement resulting from standard treatment without digital augmentation) and the progress due to the addition of the digital intervention. In essence, these two effects are intertwined and challenging to distinguish from one another.

A potentially better approach could involve a parallel group design, where children are randomly assigned to one of the two interventions, eliminating the confounding of effects, and allowing for a clearer assessment of the digital intervention's true impact.

Reviewer #2: In this study protocol, a prospective clinical trial is being proposed to determine the efficacy of adding Hear Me Read to in-person speech-language therapy for children who are deaf or hard of hearing. The primary goal of the study is to compare measures of vocabulary, speech, and language outcomes between in-person speech-language therapy alone and in-person speech-language therapy with the Hear Me Read intervention.

Major revision:

Identify all the statistical methods that will be used to conduct the specified statistical analysis.

Specific revisions:

1- Statistical Analysis section: State the type of repeated measures analysis that will be used “to evaluate the effect of intervention on these outcomes.” (p. 13).

2- Statistical Analysis section: Explain how the primary and secondary outcome measures will be evaluated. Expand upon the statement: “For primary and secondary outcome measures, changes in raw score, standard score, percentile rank, scaled score, and age equivalent for the ROWPVT-4 and CELFP3 will be evaluated.”

3- Identify the statistical method(s) that will be used to evaluate the patient characteristics associations with changes in outcomes.

4- State how the data will be summarized: means, standard deviations, medians, interquartile range, frequencies, percentages, etc.

5- To assist in the review process, add line numbering to the document.

Reviewer #3: Overall, the article details the study plans and protocol in a way that is well-reasoned, structured, and informative. The authors make a strong case for the need for an intervention such as HMR and describe plans to trial it with a study employing rigorous methodology. The use of a repeated measures design and a within-subjects approach allows for a more accurate assessment of the impact of digital intervention by eliminating potential confounding variables. This approach certainly enhances the validity of your study findings. It will be interesting to see your study population and how you might consider or account for the potential for differences in growth between Time measure 1 (baseline to post-SLT-only) and Time measure 2 (post-SLT-only to post-SLP+digital) that could be due to child-specific or provider-specific variables in Time measure 2 (increased child age, increased child-provider rapport, increased time practicing therapy techniques, etc. You may wish to discuss in this or subsequent articles.

The regular follow-up with caregivers regarding book progress and completion of weekly reading questionnaires demonstrates a proactive approach to ensuring data accuracy and participant compliance. appreciated the included details about the features of the HMR application, such as the dual interface, personalized prompts, multimodal access to books, and encouragement of dialogic reading strategies were commendable. I would like to see more discussion related to the intervention design, such as how and why features and/or strategies were chosen to be part of the HMR intervention, including rationale from existing evidence, literature, other digital storybook interventions, etc. A description or examples of the expectations of use for participants would be helpful as well for readers to gain understanding of the application and the researchers’ intents for use as an effective intervention tool.

While the article mentions that access to interim analyses and the dataset is restricted to authorized entities, it does not provide clear justification for this limitation. As written in the PLOS data policy, authors are required to make all data fully available as part of the manuscript or deposited to a public repository. The lack of public availability of the dataset hinders potential replication and further scrutiny of the study findings.

7. PLOS authors have the option to publish the peer review history of their article (what does this mean?). If published, this will include your full peer review and any attached files.

Reviewer #1: No

Reviewer #2: No

Reviewer #3: **Yes: **Amanda Rudge

---

## [Author Response · Author response to Decision Letter 0]

19 Jan 2024

Reviewer 1 Comments to the Author: 

Comment: This particular pre-post design is suitable when one assumes that the underlying mechanism, in this case, the improvement in children's speech and language skills, remains relatively consistent or stable over time. It is reasonable to assume that providing standard therapy over the 12-month period alone should lead to some level of improvement in the measured outcomes. If that is indeed the case, the difference between the two conditions (SLT-only during the first six months and SLT+Digital during the last six months) will encompass both the impact of 'natural progress' (i.e., improvement resulting from standard treatment without digital augmentation) and the progress due to the addition of the digital intervention. In essence, these two effects are intertwined and challenging to distinguish from one another.

A potentially better approach could involve a parallel group design, where children are randomly assigned to one of the two interventions, eliminating the confounding of effects, and allowing for a clearer assessment of the digital intervention's true impact.

Response: Thank you for your comment. A parallel group design insinuates that children would be randomized into one of two groups (clinical care as usual or clinical care + HMR intervention). While this would be ideal, this design does not account for the differences in therapy frequency. These differences range from weekly in-person speech therapy to evaluation-only speech therapy every six months. Additionally, this is a pilot trial with a small sample size (N=50). Using a within-subjects design allows us to achieve greater statistical power despite our small sample. Lastly, children act as their own control, which decreases variability (e.g., individual differences in therapy frequency and patient characteristics) and allows for greater confidence that changes in outcome measures are due to the addition of the Hear Me Read intervention.

Regarding natural progress: The CELF-P3 and ROWPVT-4 are standardized, norm-referenced, speech-language assessments. 

Before enrollment, children complete their Pre-Trial evaluation. About six months later, they complete the SLT-only evaluation. Lastly, they complete the SLT+Digital evaluation around 12 months of participation. Children only receive clinical recommendations (i.e., speech therapy, reading, and literacy recommendations from the speech therapist) from the Pre-Trial to SLT only evaluation. We expect to understand the child’s natural progress within this period. From the SLT only to SLT+Digital evaluation, children receive clinical recommendations and the HMR application. Residual changes in scores are expected to be attributed to the HMR application. Additionally, changes in speech, language, and literacy for children who are D/HH are most sensitive to early reading. As such, we track reading progress (using the weekly reading questionnaires) and employ standardized reading time (20 minutes per day, three times per week) throughout the study duration. 

Reviewer 2 Comments to the Author: 

Comment 1: Statistical Analysis section: State the type of repeated measures analysis that will be used “to evaluate the effect of intervention on these outcomes.” (p. 13).

Response: A repeated measures approach using a mixed effects model will be used to compare the change across time in group 1 to the change across time in group 2 (SLT-Only to SLT+Digital) [see lines 288-291 in revised manuscript]. 

Comment 2: Statistical Analysis section: Explain how the primary and secondary outcome measures will be evaluated. Expand upon the statement: “For primary and secondary outcome measures, changes in raw score, standard score, percentile rank, scaled score, and age equivalent for the ROWPVT-4 and CELFP3 will be evaluated.”

Response: Changes in raw score are reported as standard score, percentile rank, scaled score, and age equivalent for the ROWPVT-4 and CELF-P3. These changes will be evaluated given the sensitivity of raw scores [see lines 291-293 in revised manuscript]. 

Comment 3: Identify the statistical method(s) that will be used to evaluate the patient characteristics associations with changes in outcomes.

Response: We will use t-tests and Wilcoxon-Mann-Whitney tests to identify patient characteristics associated with changes in outcomes [see lines 294-295 in revised manuscript].

Comment 4: 4- State how the data will be summarized: means, standard deviations, medians, interquartile range, frequencies, percentages, etc.

Response: A table in the final manuscript will provide baseline characteristics of participants. This will include means (SD), medians (IQR), and frequencies, as appropriate given the baseline variable of interest. Outcome measures will report the change in means from pre-trial to SLT only, from SLT only to SLT+Digital, and from pre-trial to SLT+Digital.

Comment 5: To assist in the review process, add line numbering to the document.

Response: Line numbers were added to the document.

Reviewer 3 Comments to the Author: 

Comment: Overall, the article details the study plans and protocol in a way that is well-reasoned, structured, and informative. The authors make a strong case for the need for an intervention such as HMR and describe plans to trial it with a study employing rigorous methodology. The use of a repeated measures design and a within-subjects approach allows for a more accurate assessment of the impact of digital intervention by eliminating potential confounding variables. This approach certainly enhances the validity of your study findings. It will be interesting to see your study population and how you might consider or account for the potential for differences in growth between Time measure 1 (baseline to post-SLT-only) and Time measure 2 (post-SLT-only to post-SLP+digital) that could be due to child-specific or provider-specific variables in Time measure 2 (increased child age, increased child-provider rapport, increased time practicing therapy techniques, etc. You may wish to discuss in this or subsequent articles.

Response: Thank you for your comment. Child-, caregiver-, and provider-specific variables are important to consider. We will use t-tests and Wilcoxon-Mann-Whitney tests to identify patient characteristics associated with changes in outcomes. If any imbalance in patient characteristics (e.g., age, frequency of therapy) is detected between groups, the imbalanced characteristics will be adjusted for in a repeated measures approach using a mixed effects model [see lines 294-298 in the revised manuscript]. We expect parent/provider-specific variables to impact the child’s progress/outcomes equally during both study periods (Pre-Trial to SLT only and SLT only to SLT+Digital). Standard clinical care should minimize variability in how SLPs conduct speech therapy and administer speech evaluations. Regarding parent-specific variables, parents are instructed to read with their child 20 minutes per day, three times per week. Should differences in child-clinician rapport, parent availability to read with the child, etc., warrant investigating the effects of parent/provider-specific variables, we will discuss this in subsequent articles.

Comment: The regular follow-up with caregivers regarding book progress and completion of weekly reading questionnaires demonstrates a proactive approach to ensuring data accuracy and participant compliance. appreciated the included details about the features of the HMR application, such as the dual interface, personalized prompts, multimodal access to books, and encouragement of dialogic reading strategies were commendable. I would like to see more discussion related to the intervention design, such as how and why features and/or strategies were chosen to be part of the HMR intervention, including rationale from existing evidence, literature, other digital storybook interventions, etc. A description or examples of the expectations of use for participants would be helpful as well for readers to gain understanding of the application and the researchers’ intents for use as an effective intervention tool.

Response: Thank you for your comment. We conducted a preliminary usability assessment and employed qualitative interview techniques to understand the needs of children who are D/HH and their caregivers. This usability assessment found that families had default, specific, and family needs as it relates to the application. We developed the HMR application in line with these needs [see lines 204-215 in revised manuscript for more details].

Comment: While the article mentions that access to interim analyses and the dataset is restricted to authorized entities, it does not provide clear justification for this limitation. As written in the PLOS data policy, authors are required to make all data fully available as part of the manuscript or deposited to a public repository. The lack of public availability of the dataset hinders potential replication and further scrutiny of the study findings.

Response: Thank you for your comment. Given our small sample size (N=50), interim analyses are not planned. This prospective clinical trial will serve as pilot data to help us determine outcome measures for future trials. Anticipated completion of primary analyses is September 2024, at which point, result of outcome measures will be made available on ClinicalTrials.gov in line with their requirements. Additionally, see lines 299-300 in revised manuscript for updated language about data availability.

---

## [Decision Letter · Decision Letter 1]

13 Feb 2024

PONE-D-23-28173R1Study protocol for Hear Me Read (HMR): A prospective clinical trial assessing a digital storybook intervention for young children who are deaf or hard of hearingPLOS ONE

Dear Dr. Malhotra,

Thank you for submitting your manuscript to PLOS ONE. After careful consideration, we feel that it has merit but does not fully meet PLOS ONE’s publication criteria as it currently stands. Therefore, we invite you to submit a revised version of the manuscript that addresses the points raised during the review process.

We look forward to receiving your revised manuscript.

Kind regards,

Renato S. Melo, PhD

Academic Editor

PLOS ONE

Journal Requirements:

Reviewers' comments:

Reviewer's Responses to Questions

**Comments to the Author**

1. Does the manuscript provide a valid rationale for the proposed study, with clearly identified and justified research questions?

Reviewer #2: Yes

Reviewer #3: Yes

2. Is the protocol technically sound and planned in a manner that will lead to a meaningful outcome and allow testing the stated hypotheses?

Reviewer #2: Yes

Reviewer #3: Yes

3. Is the methodology feasible and described in sufficient detail to allow the work to be replicable?

Reviewer #2: Yes

Reviewer #3: Yes

4. Have the authors described where all data underlying the findings will be made available when the study is complete?

Reviewer #2: No

Reviewer #3: Yes

5. Is the manuscript presented in an intelligible fashion and written in standard English?

Reviewer #2: Yes

Reviewer #3: Yes

6. Review Comments to the Author

You may also provide optional suggestions and comments to authors that they might find helpful in planning their study.

Reviewer #2: Minor Revisions:

Overall, the statistical methods require further clarification.

1- Line 220: Alpha levels rather than p-values are used in sample size calculations. Typically alpha levels are in the range of 0.01 to 0.10. Justify an alpha level of 0.001. It seems too small. Perhaps this is a typographical error.

2- Line 289: In the mixed effects model, it seems appropriate to first test the interaction of group by time.

3- Typically continuous baseline characteristics/demographics are compared using t-tests or Wilcoxon rank sum tests and categorical data is compared between two groups using chi-square tests. T-tests and Wilcoxon Mann-Whitney U tests do not test for association, but differences between 2 groups.

4- Line 295: The following statement is vague. Is the analytic approach mixed linear modeling? Clarify. "Characteristics associated with changes in the outcome (e.g., patient characteristics such as age, frequency of therapy) will be adjusted for within the context of the aforementioned analytic approach."

Reviewer #3: I am satisfied with the authors' responses to concerns and recommendations raised in the initial review. Authors have provided needed clarity surrounding the development of the HMR application, planned analyses, and data availability. These revisions have strengthened the manuscript. The work is a valuable contribution to the field.

7. PLOS authors have the option to publish the peer review history of their article (what does this mean?). If published, this will include your full peer review and any attached files.

Reviewer #2: No

Reviewer #3: No

---

## [Author Response · Author response to Decision Letter 1]

22 Mar 2024

Journal Requirements: 

Response: We have reviewed the reference list and identified no retracted articles. In response to Review 2 comments, we added the following to the reference list: (27) Kohn MA, Senyak J. Sample Size Calculators [website]. UCSF CTSI. 14 March 2024. Available at https://www.sample-size.net/ [Accessed 19 March 2024].

Reviewer 2 Comments to the Author: 

Comment 1: Line 220: Alpha levels rather than p-values are used in sample size calculations. Typically alpha levels are in the range of 0.01 to 0.10. Justify an alpha level of 0.001. It seems too small. Perhaps this is a typographical error.

Response 1: Thank you for pointing out this error. It appears that in addition to the typographical error of putting p instead of alpha, there was an error made in calculation by using alpha=0.001%, or 0.00001, instead of alpha=0.001. This caused the sample size estimate to be n=43, when the correct calculation of the given parameters gives n=27. This has been corrected in the submission [see lines 220-226, source cite in reference list (27)]. With regards to the alpha level, alpha=0.001 was chosen to be conservative and ensure that the treatment really was effective before expanding to larger studies, especially as HMR poses very low risk to patients. Using the same parameters but changing alpha level to 0.01 and 0.05 gives us sample sizes of n=19 and n=13, respectively, but for the purposes of our study we believe the feasibility assessment to be more important as we seek to gain as much information as possible.

Comment 2: Line 289: In the mixed effects model, it seems appropriate to first test the interaction of group by time.

Response 2: That is correct. We are comparing the changes between the two intervention periods (the SLT-Only and the SLT+Digital) which is a group by time interaction. 

Comment 3: Typically continuous baseline characteristics/demographics are compared using t-tests or Wilcoxon rank sum tests and categorical data is compared between two groups using chi-square tests. T-tests and Wilcoxon Mann-Whitney U tests do not test for association, but differences between 2 groups.

Response 3: Thank you for your comment. We have included updated language as follows: “Baseline assessments are used to identify patient characteristics (e.g., age, frequency of therapy, rapport with SLP) associated with outcome variables using t-tests and correlation analyses.” [lines 293-295].

Comment 4: Line 295: The following statement is vague. Is the analytic approach mixed linear modeling? Clarify. "Characteristics associated with changes in the outcome (e.g., patient characteristics such as age, frequency of therapy) will be adjusted for within the context of the aforementioned analytic approach."

Response 4: We have clarified the statement as follows: “Characteristics associated with changes in the outcome (e.g., patient characteristics such as age, frequency of therapy) will be adjusted for within the context of a repeated measures approach using a mixed effects model” [see lines 296-298].

---

## [Decision Letter · Decision Letter 2]

11 Apr 2024

Study protocol for Hear Me Read (HMR): A prospective clinical trial assessing a digital storybook intervention for young children who are deaf or hard of hearing

PONE-D-23-28173R2

Dear Dr. Malhotra,

We’re pleased to inform you that your manuscript has been judged scientifically suitable for publication and will be formally accepted for publication once it meets all outstanding technical requirements.

Kind regards,

Renato S. Melo, PhD

Academic Editor

PLOS ONE

Additional Editor Comments (optional):

Reviewers' comments:

Reviewer's Responses to Questions

**Comments to the Author**

1. Does the manuscript provide a valid rationale for the proposed study, with clearly identified and justified research questions?

Reviewer #2: Yes

Reviewer #3: Yes

2. Is the protocol technically sound and planned in a manner that will lead to a meaningful outcome and allow testing the stated hypotheses?

Reviewer #2: Yes

Reviewer #3: Yes

3. Is the methodology feasible and described in sufficient detail to allow the work to be replicable?

Reviewer #2: Yes

Reviewer #3: Yes

4. Have the authors described where all data underlying the findings will be made available when the study is complete?

Reviewer #2: No

Reviewer #3: Yes

5. Is the manuscript presented in an intelligible fashion and written in standard English?

Reviewer #2: Yes

Reviewer #3: Yes

6. Review Comments to the Author

You may also provide optional suggestions and comments to authors that they might find helpful in planning their study.

Reviewer #2: All comments have been adequately addressed.

Reviewer #3: I am satisfied with the authors' responses to concerns and recommendations raised in the initial and revision 1 reviews. These revisions have strengthened the manuscript. The work is a valuable contribution to the field.

7. PLOS authors have the option to publish the peer review history of their article (what does this mean?). If published, this will include your full peer review and any attached files.

Reviewer #2: No

Reviewer #3: No

---

## [Editor Report · Acceptance letter]

2 May 2024

PONE-D-23-28173R2 

PLOS ONE

Dear Dr. Malhotra, 

I'm pleased to inform you that your manuscript has been deemed suitable for publication in PLOS ONE. Congratulations! Your manuscript is now being handed over to our production team.

Kind regards, 

on behalf of

Dr. Renato S. Melo 

Academic Editor

PLOS ONE